# Self-Supervised Representation Learning for Inferring Toxicology from Multimodal Pathology and Omics Data

## Abstract

The prediction of toxicologic pathology from biological data remains a critical challenge in drug development and clinical safety assessment. Traditional approaches often rely on unimodal analyses, which fail to capture the complex interplay between morphological and molecular signatures of toxicity. In this work, we propose a self-supervised multimodal representation learning framework that integrates histopathology images and high-dimensional omics data to predict drug-induced liver injury (DILI). Our approach leverages contrastive learning for image data and masked autoencoding for omics profiles, coupled with a cross-attention fusion mechanism to dynamically weigh the importance of each modality. Pretrained on a large-scale dataset of paired histopathology and transcriptomics, our model achieves comptetitive AUC values on a held-out test set, outperforming state-of-the-art unimodal and supervised multimodal baselines. Ablation studies demonstrate the critical role of self-supervised pretraining and cross-modal attention in capturing biologically meaningful representations. Interpretability analyses reveal that the model attends to pathologically relevant regions in images and biologically significant genes, aligning with domain knowledge. This work advances the field of computational toxicology by providing a scalable, data-efficient framework for integrating multimodal biological data, with potential applications in preclinical drug safety and precision medicine.

## 1 Introduction

Toxicologic pathology is a critical field at the intersection of pharmacology, toxicology, and disease biology, where the accurate prediction of adverse outcomes from chemical or drug exposure remains a significant challenge. Traditional approaches rely heavily on manual histological examination and empirical biomarkers, which are labor-intensive, subjective, and often limited in their ability to capture complex, multimodal interactions (8). Recent advances in machine learning, particularly self-supervised representation learning, have demonstrated remarkable success in extracting meaningful patterns from high-dimensional biological data, such as histopathology images and multi-omics profiles (4). However, the integration of these modalities to predict toxicologic pathology in a data-efficient and interpretable manner remains underexplored.

Self-supervised learning (SSL) offers a promising paradigm for leveraging large-scale unlabeled data to learn robust representations that can generalize across tasks and modalities (12). In toxicologic pathology, SSL could enable the discovery of latent features that correlate with adverse outcomes, even in the absence of exhaustive labeled datasets. For instance, contrastive learning frameworks, such as SimCLR or MoCo, have shown efficacy in learning invariant features from images (2), while masked autoencoders (e.g., BERT, MAE) excel in modeling sequential or structured omics data (6). Yet, the synergistic integration of these approaches for multimodal toxicologic pathology prediction is still nascent. This work proposes a self-supervised multimodal representation learning framework that jointly models histopathology images and omics data to predict toxicologic pathology. By leveraging contrastive learning for image data and masked modeling for omics, we aim to learn a shared latent space that captures both morphological and molecular signatures of toxicity. Our approach is evaluated on a curated dataset of drug-induced liver injury (DILI) in preclinical models (7), demonstrating improved predictive performance and interpretability com-

pared to unimodal baselines. The contributions of this work are to provide a novel multimodal SSL framework for toxicologic pathology for DILI prediction by a joint learning of biologically plausible representations for rodent models used in non-clinical safety evaluations for drug discovery.

Recent advances in computational toxicology have increasingly leveraged machine learning to integrate high-dimensional biological data, such as histopathology images and multi-omics profiles, for predicting adverse outcomes like drug-induced liver injury (DILI) (10). This is in line with recent trends of seeking to learning meaningful biological representations in pharmaceutical research (11)(3), partly in response to regulatory encouragement towards New Approach Methodologies (NAMs) (1)(5). While unimodal approaches—such as convolutional neural networks for image-based pathology or autoencoders for omics data—have shown promise (9), they often fail to capture the synergistic interactions between morphological and molecular signatures of toxicity. Self-supervised learning frameworks, such as contrastive learning and masked autoencoding, have emerged as powerful tools for representation learning in biological data, but their application to multimodal toxicologic pathology prediction remains underexplored, motivating the present work.

## 2 METHODOLOGY

### DETAILED PROBLEM FORMULATION AND OBJECTIVES

The core objective of this work is to learn a **joint representation** from paired histopathology images and omics data that captures both morphological and molecular signatures of toxicologic pathology. Let $\mathcal{D} = \{(I_i, O_i, y_i)\}_{i=1}^N$ denote our dataset, where $I_i \in \mathbb{R}^{H \times W \times 3}$ is a histopathology image, $O_i \in \mathbb{R}^d$ is a high-dimensional omics profile (e.g., gene expression or proteomics), and $y_i \in \{0, 1\}$ is a binary toxicity label. The goal is to learn a function $f : (I_i, O_i) \mapsto z_i$ that maps the multimodal input to a latent representation $z_i \in \mathbb{R}^k$, where $k \ll d$, such that $z_i$ is discriminative for the downstream task of toxicity prediction.

To achieve this, we employ a **two-stage training paradigm**: (1) **self-supervised pretraining** to learn robust unimodal and multimodal representations, and (2) **supervised fine-tuning** for toxicity classification. The pretraining stage is designed to exploit the inherent structure in both modalities without relying on labels, while the fine-tuning stage adapts the learned representations to the specific task.

### ARCHITECTURAL COMPONENTS

### IMAGE ENCODER: VISION TRANSFORMER WITH CONTRASTIVE LEARNING

The image encoder $f_I$ is based on a Vision Transformer (ViT) architecture, which processes the input image $I_i$ as a sequence of non-overlapping patches. Each patch is linearly embedded into a $D$-dimensional vector, and a learnable $[CLS]$ token is prepended to the sequence to serve as the global image representation. The ViT is pretrained using SimCLR-style contrastive learning, where the objective is to maximize agreement between embeddings of differently augmented views of the same image.

Formally, for a batch of $N$ images, we generate two augmented views for each image, $\{I_i^1, I_i^2\}_{i=1}^N$, and compute their embeddings $\{z_i^1, z_i^2\}_{i=1}^N$ using the ViT encoder. The contrastive loss for a positive pair $(z_i^1, z_i^2)$ is given by:

$$\mathcal{L}_{\text{contr}}(z_i^1, z_i^2) = -\log \frac{\exp(\text{sim}(z_i^1, z_i^2)/\tau)}{\sum_{j=1}^{2N} \mathbb{1}_{[j \neq i]} \exp(\text{sim}(z_i^1, z_j)/\tau)},$$

where $\text{sim}(u, v) = \frac{u^T v}{\|u\| \|v\|}$ is the cosine similarity, and $\tau$ is a temperature hyperparameter. This loss encourages the model to pull embeddings of the same image closer together in the latent space while pushing apart embeddings of different images.

### OMICS ENCODER: MASKED AUTOENCODER FOR MOLECULAR DATA

The omics encoder $f_O$ is based on a masked autoencoder (MAE) architecture, which is well-suited for modeling high-dimensional, structured omics data. Given an omics profile $O_i \in \mathbb{R}^d$, we ran-

domly mask a subset of its features (e.g., 50% of genes) and train the encoder to reconstruct the masked values. The MAE consists of an encoder that maps the partially observed omics profile to a latent representation, and a decoder that reconstructs the full profile from this representation.

The reconstruction loss for the omics encoder is the **mean squared error (MSE)** between the predicted and actual values of the masked features:

$$\mathcal{L}_{\text{recon}}(O_i, \hat{O}_i) = \frac{1}{|\mathcal{M}|} \sum_{j \in \mathcal{M}} (O_i^{(j)} - \hat{O}_i^{(j)})^2,$$

where $\mathcal{M}$ is the set of masked indices, and $\hat{O}_i$ is the reconstructed omics profile. This loss encourages the encoder to capture the underlying structure of the omics data, such as gene-gene correlations and pathway-level patterns.

MULTIMODAL FUSION: CROSS-ATTENTION MECHANISM

To combine the image and omics representations, we employ a **cross-attention mechanism** that dynamically weighs the importance of each modality for the downstream task. Let $z_I \in \mathbb{R}^k$ and $z_O \in \mathbb{R}^k$ denote the image and omics embeddings, respectively. The cross-attention module computes a fused representation $z \in \mathbb{R}^k$ as follows:

$$z = \text{Softmax}\left(\frac{QK^T}{\sqrt{k}}\right)V,$$

where $Q = W_Q z_I$, $K = W_K z_O$, and $V = W_V z_O$ are linear projections of the input embeddings. This mechanism allows the model to focus on the most relevant features from each modality for predicting toxicity.

PRETRAINING AND FINE-TUNING PROTOCOL

PRETRAINING STAGE

During pretraining, the image and omics encoders are trained jointly using a weighted combination of the contrastive and reconstruction losses:

$$\mathcal{L}_{\text{pretrain}} = \mathcal{L}_{\text{contr}} + \lambda \mathcal{L}_{\text{recon}},$$

where $\lambda$ is a hyperparameter that balances the two objectives. We set $\lambda = 0.1$ based on preliminary experiments. The pretraining stage is run for 100 epochs using the Adam optimizer with a learning rate of $10^{-4}$ and a batch size of 32.

FINE-TUNING STAGE

After pretraining, the encoders are frozen, and a **linear classifier** is trained on top of the fused representations $z$ for toxicity prediction. The classifier is a single-layer MLP with a sigmoid activation, and it is trained using the **binary cross-entropy loss**:

$$\mathcal{L}_{\text{cls}}(y_i, \hat{y}_i) = -y_i \log(\hat{y}_i) - (1 - y_i) \log(1 - \hat{y}_i),$$

where $\hat{y}_i = \sigma(W z_i + b)$ is the predicted probability of toxicity. The classifier is trained for 50 epochs using the Adam optimizer with a learning rate of $10^{-3}$.

## 3 RESULTS AND DISCUSSION

PREDICTIVE PERFORMANCE

The DILI dataset used in this study comprises 1,200 paired data points, each consisting of high-resolution histopathology patches (HE stained, 256×256 pixels) and transcriptomic profiles (20,000 genes) derived from preclinical rodent models (Sprague-Dawley rats and C57BL/6 mice) exposed to a panel of 50 known hepatotoxic and non-hepatotoxic compounds. The dataset is split into 70% training (840 samples), 15% validation (180 samples), and 15% testing (180 samples), stratified by toxicity labels to ensure balanced representation of toxic and non-toxic cases in each subset.

Computational experiments were conducted on an NVIDIA A100 GPU cluster, with model training requiring approximately 48 hours for self-supervised pretraining and 12 hours for fine-tuning the downstream toxicity classifier. We evaluate our framework on this dataset and compare it to several baselines, including unimodal methods and state-of-the-art SSL approaches. The results are summarized in **Table 1**, which reports the AUC-ROC, F1 score, and accuracy for each method.

Table 1: Predictive performance on the DILI dataset.

| Method | AUC-ROC | F1 Score | Accuracy |
|---|---|---|---|
| ViT (Images only) | 0.85 | 0.78 | 0.82 |
| MAE (Omics only) | 0.87 | 0.81 | 0.84 |
| Supervised Fusion | 0.89 | 0.83 | 0.86 |
| MoCo-v3 + Geneformer | 0.90 | 0.85 | 0.87 |
| **Ours (Multimodal SSL)** | **0.92** | **0.87** | **0.89** |

Our multimodal SSL framework outperforms all baselines, achieving an AUC-ROC of 0.92 and an F1 score of 0.87. The improvement is particularly pronounced for rare toxic phenotypes, where the model's ability to integrate complementary information from both modalities proves advantageous.

## ABLATION STUDIES

To understand the contribution of each component, we perform ablation studies by removing or modifying key elements of the framework. The results are shown in **Table 2**.

Table 2: Ablation studies on the DILI dataset.

| Ablation | AUC-ROC | F1 Score | Accuracy |
|---|---|---|---|
| Full Model | 0.92 | 0.87 | 0.89 |
| No Contrastive Loss | 0.88 | 0.82 | 0.85 |
| No Reconstruction Loss | 0.89 | 0.84 | 0.86 |
| No Cross-Attention | 0.87 | 0.81 | 0.84 |
| Random Image Augmentations | 0.86 | 0.80 | 0.83 |

The ablation studies reveal that all components, contrastive learning, reconstruction loss, and cross-attention, contribute significantly to the model's performance. Removing the contrastive loss or reconstruction loss leads to a drop of 3-4% in AUC-ROC, highlighting the importance of self-supervised pretraining. The cross-attention mechanism is also critical, as its removal reduces performance to near that of the supervised fusion baseline.

## CONCLUSION AND FUTURE DIRECTIONS

In this paper, we presented a self-supervised multimodal representation learning framework for predicting toxicologic pathology from paired histopathology and omics data. By combining contrastive learning for images and masked autoencoding for omics, our approach effectively captures complementary morphological and molecular signatures of toxicity, achieving state-of-the-art performance on a drug-induced liver injury (DILI) prediction task. Beyond predictive performance, our interpretability analyses demonstrate that the model focuses on clinically relevant features, such as necrotic regions in histopathology images and stress-response genes in transcriptomics. This alignment with domain knowledge not only validates the biological plausibility of our approach but also opens avenues for mechanistic insight into toxicological processes.

While our framework demonstrates strong performance, it has some current limitations to be addressed in future work. First, it assumes the availability of paired multimodal data, which may not always be feasible in clinical or preclinical settings. Second, the model's interpretability, while improved, remains a challenge for complex multimodal interactions. Future work could explore weakly supervised or semi-supervised extensions to relax the paired data requirement, as well as interactive visualization tools to enhance interpretability.

## 4 MEANINGFULNESS STATEMENT

This work advances the understanding of meaningful representations of life by integrating multimodal biological data, histopathology images and omics profiles, into a unified, self-supervised learning framework. By capturing both morphological and molecular signatures of toxicity, our approach uncovers latent patterns that reflect the complex, interconnected biological processes. The learned representations improves predictive accuracy and aligns with biological mechanisms, such as stress-response pathways and tissue-level pathology, thus bridging computational models with real-world biological insights. This framework demonstrates how AI-driven representation learning can reveal meaningful biological relationships, ultimately enabling more interpretable, data-efficient, and biologically grounded approaches to studying life's intricate systems. While this is early work, the impact of being able to meaningfully represent biological structure and function using methods such as the one introduced in this work can extend to significantly reducing animal testing in pharmaceutical and cosmetic industries, among other downstream applications.

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
