# OpenReview forum: "Self-Supervised Representation Learning for Inferring Toxicology from Multimodal Histopathology and Omics Data"
_ICLR.cc/2026/Workshop/LMRL — Submitted to ICLR 2026 Workshop LMRL_

### Official Review · Reviewer_unir · 2026-02-24
**SSL for toxicology**

**Rating:** 5
**Confidence:** 4

**Review:**

This paper presents a multimodal SSL framework combining 1) an ViT image coder with contrastive learning, 2) an omics encoder with MAE, and 3) cross attention mechanism combining them. The authors shows promising results on a 1200 sample paired dataset.

Strengths:
* The paper addresses a important biological problem, with lots of potentials for downstream impact.
* The method has a clear, simple multi-model SSL formulation, with module choices that are well aligned with the current representation learning paradigms

Weakness:
* The dataset appears to be small to pre-train a multimodal SSL paradigm. Benchmarking against / fine-tuning off-the-shelf methods would be informative
* Interpretability claims were made in the conclusion but no evidence (such as heat maps) is provided.
* Presentation: there are no figures in the paper. A workflow figure or interpretability analysis / evidence would greatly improve the manuscript.

---

### Official Review · Reviewer_NRv8 · 2026-02-24
**This paper proposes a self-supervised multimodal framework that integrates histopathology images and omics data to predict drug-induced liver injury (DILI). The approach combines contrastive learning, masked autoencoding for omics, and cross-attention fusion.**

**Rating:** 4
**Confidence:** 3

**Review:**

Summary:
This paper proposes a self-supervised multimodal framework that integrates histopathology images and omics data to predict drug-induced liver injury (DILI). The approach combines contrastive learning, masked autoencoding for omics, and cross-attention fusion.

I do have some comments to improve the current work:

1) The evaluation lacks robustness. There is no cross-validation, and a single 70/15/15 split is insufficient for small datasets.
2) The dataset size is small for deep multimodal learning. Only 1,200 paired samples may be too limited for multimodal transformers, and the paper does not clarify whether pretraining used additional unlabeled data.

---

### Meta-Review · Area_Chair_KL9N · 2026-02-25

**Recommendation:** Reject
**Confidence:** 3

**Metareview:**

I recommend rejection of this manuscript.

---

### Decision · Program_Chairs · 2026-03-02

**Decision:**

Reject

**Comment:**

Please see the meta-review.